# Behavioral thermoregulation by reptile embryos promotes hatching success and synchronization

Shuo Liu[1,2], Bo Zhao[3], Xiaoting Gu [2,4] & Weiguo Du [1✉]

Reptile embryos can move inside eggs to seek optimal thermal conditions, falsifying the traditional assumption that embryos are simply passive occupants within their eggs. However, the adaptive significance of this thermoregulatory behavior remains a contentious topic. Here we demonstrate that behavioral thermoregulation by turtle embryos shortened incubation periods which may reduce the duration of exposure to dangerous environments, decreased egg mortality imposed by lethally high temperatures, and synchronized hatching which reduces predation risk. Our study provides empirical evidence that behavioral thermoregulation by turtle embryos is adaptive.

[1] Key Laboratory of Animal Ecology and Conservation Biology, Institute of Zoology, Chinese Academy of Sciences, Beijing 100101, China. [2] College of life sciences, University of Chinese Academy of Sciences, Beijing 100101, China. [3] College of Fisheries, Zhejiang Ocean University, Zhoushan 316000, China. [4] State Key Laboratory of Integrated Management of Pest Insects and Rodents, Institute of Zoology, Chinese Academy of Sciences, Beijing 100101, China. ✉email: duweiguo@ioz.ac.cn

Although the behavior of post-embryonic life stages is well documented, the complexity and adaptive significance of embryonic behavior has been underestimated and understudied[1–3]. Embryos were traditionally assumed to be passive to their environments, but increasing evidence shows that embryos are able to adjust behaviorally and physiologically in response to environmental changes[1]. One impressive case of embryonic behavior is the observation that embryos of oviparous reptiles can move inside their eggs to seek suitable thermal environments[4–6]. Nonetheless, the adaptive significance of embryonic thermoregulatory behavior remains hotly disputed[7,8]; some authors claim that embryonic thermoregulatory behavior improves offspring viability[4,7] whereas others argue that this behavior is selectively neutral, given limited thermal heterogeneity within eggs and the limited ability of embryos to move within the egg[8,9]. Actually, embryonic behavioral thermoregulation may be possible in relatively large eggs (e.g., turtles), but not in small eggs (e.g., lizards)[6].

If embryonic behavioral thermoregulation is adaptive, this behavior should enable an embryo to find a thermal environment that improves its fitness in light of the cost-benefit model of thermoregulation[10]. Theoretically, behavioral thermoregulation by embryos may improve developmental success via the following pathways[4,11]. First, behavioral thermoregulation may enable embryos to find thermal environments that accelerate development. Second, behavioral thermoregulation may reduce thermal differentials among eggs within a nest, which synchronizes hatching to dilute predation risk. Third, behavioral thermoregulation may enable embryos to avoid lethal thermal extremes, and thereby reduce embryonic mortality during development. To test these hypotheses, we inhibited thermoregulatory behavior of embryos in semi-natural nests by pharmacologically blocking transient receptor potential channels (TRPs) that sense temperature changes[12], in order to measure the effects of embryonic behavioral thermoregulation on incubation period, hatching success, and hatching synchronization in the Chinese softshell turtle (*Pelodiscus sinensis*). By doing so, we have been able to document the adaptive significance of embryonic behavioral thermoregulation in the species where we first discovered this behavioral phenomenon.

## Results

**Effects of capsazepine on embryonic development and hatchling traits.** We used capsazepine to inhibit thermoregulatory behavior of turtle embryos[11]. To test whether capsazepine itself affects embryonic development and hatchling traits, we conducted egg-incubation experiments in the laboratory at a constant temperature of 30 °C; thus, there was no thermal gradient available for behavioral thermoregulation by embryos. The application of capsazepine on eggs did not affect incubation period, variation of incubation period, hatching success, or hatchling traits including carapace length and width, body mass, and righting time (Table S1). In addition, the uniform hatching date among eggs suggested synchronous hatching in this species (Table S1). Therefore, capsazepine did not affect embryonic development or hatchling traits.

**Effects of behavioral thermoregulation on embryonic development and hatchling traits.** To identify effects of behavioral thermoregulation on embryonic development and hatchling traits, we manipulated the thermoregulatory behavior of turtle embryos undergoing development in semi-natural nests that we constructed in sunlight-exposed (open), filtered, or fully-shaded habitats. We monitored the developmental stages of embryos during the experiment. We found no between-group differences in developmental stages at weeks 2–4 (week 2, $Z = -1.533$, $p = 0.125$; week 3, $Z = -0.238$, $p = 0.812$; week 4, $Z = -0.692$, $p = 0.489$). However, developmental stages were significantly advanced in the thermoregulation group compared to the thermoregulation-inhibited group at weeks 5–7 (week 5, $Z = -2.175$, $p = 0.030$; week 6, $Z = -2.678$, $p = 0.007$; week 7, $Z = 2.754$, $p = 0.006$; Fig. 1).

The thermoregulation-inhibited group had a longer mean incubation period than did the thermoregulation group ($58.06 \pm 0.15$ days versus $54.91 \pm 0.07$ days; $F_{1,416} = 356.995$, $p < 0.001$; Fig. 2a). In addition, incubation periods of the thermoregulation group exhibited lower variation than did those of the thermoregulation-inhibited group in terms of the coefficient of variation ($F_{1,87} = 65.680$, $p < 0.001$; Fig. 2b) and the range of variation ($F_{1,88} = 66.480$, $p < 0.001$; Fig. 2c). Eggs from the thermoregulation group also had higher hatching

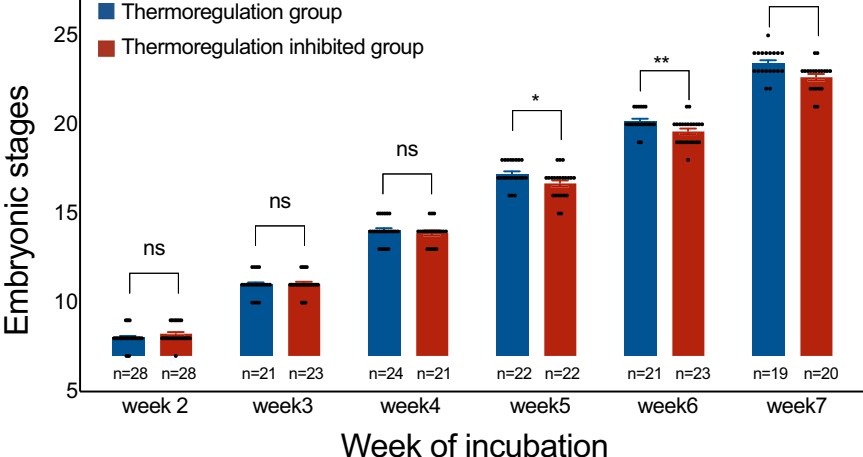

**Fig. 1 The developmental stages of embryos from the thermoregulation and thermoregulation-inhibited groups throughout the incubation period.** From week 2–4, both groups had similar developmental stages. From week 5–7, however, developmental stages advanced faster in the thermoregulation group than the thermoregulation-inhibited group. Mann-Whitney U-test was used to test the differences in the developmental stages of embryos under different treatments, Data show as mean ± SE.

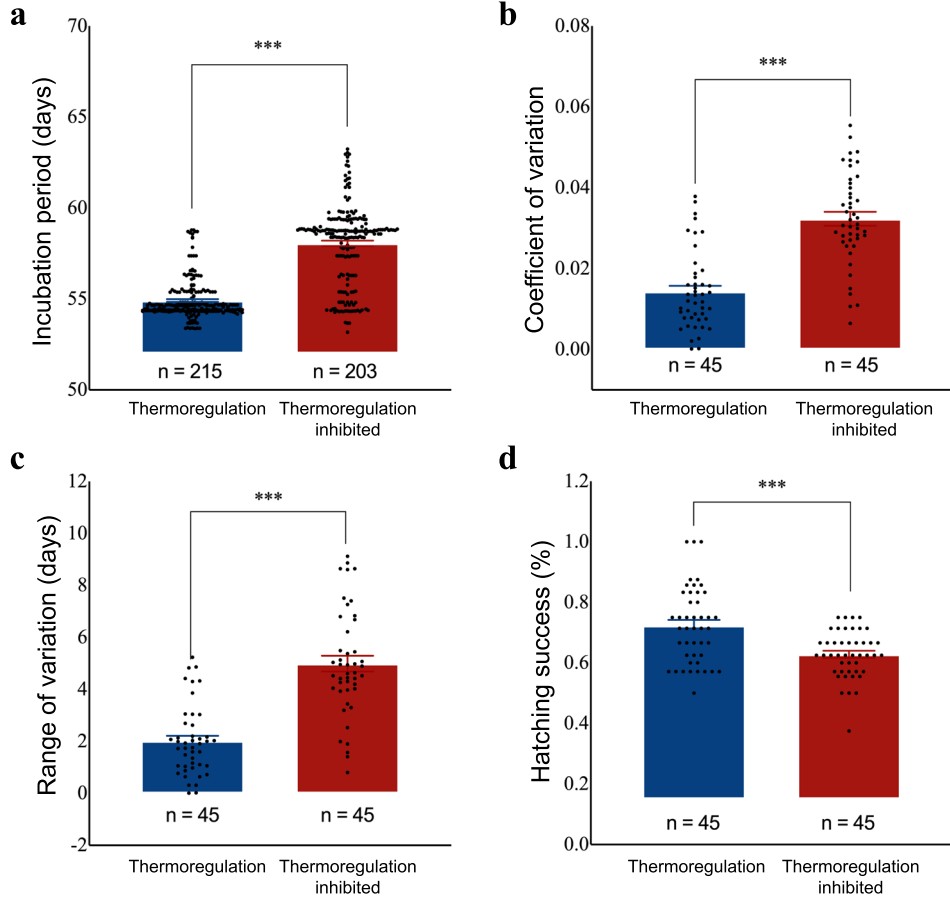

**Fig. 2 Effects of behavioral thermoregulation on incubation period and hatching success of eggs. a** Eggs from the thermoregulation group had shorter incubation periods than those from the thermoregulation-inhibited group. **b** Incubation periods had a smaller coefficient of variation in the thermoregulation group than in the thermoregulation-inhibited group. **c** Incubation periods had a narrower range of variation in the thermoregulation group than in the thermoregulation-inhibited group. **d** Eggs from the thermoregulation group had higher hatching success than those from the thermoregulation-inhibited group. Data were analyzed by generalized linear mixed models with the clutch number as the random factor. Data are presented with means ± SE. ***$p < 0.001$.

success than those from the thermoregulation-inhibited group ($F_{1,88} = 19.337$, $p < 0.001$; Fig. 2d). However, body mass ($F_{1,370} = 0.251$, $p = 0.617$) and righting time ($F_{1,374} = 0.711$, $p = 0.400$) of hatchlings did not differ between groups (Table S2).

In summary, embryonic behavioral thermoregulation shortened incubation periods, facilitated synchronous hatching, and increased hatching success. These conclusions were valid for embryos from open, filtered, or fully-shaded habitats (Table 1), despite differences in thermal environments and nest temperatures among the three habitats (Tables S3, S4).

## Discussion

By behaviorally thermoregulating, turtle embryos shortened their incubation period by accelerating their developmental rate (as shown by the advanced developmental stages; Fig. 1), as opposed to premature hatching (hatching at an earlier developmental stage) that may incur costs (e.g., reduced self-righting ability) on hatchlings[13]. Shortening incubation periods may reduce the mortality risk of embryos, given that most turtles nest beside water bodies where embryos are exposed to risks of extreme temperatures, flooding, bacterial contamination and nest predation. Reduced incubation periods leading to early hatching were found to have a positive effect on offspring growth and survival in jacky dragon lizards (*Amphibolurus muricatus*)[14], but such fitness benefits of early hatching are unknown in turtles.

Synchronous hatching occurs in many oviparous taxa from invertebrates to birds. In turtles, synchronous hatching may facilitate group emergence to dilute predation risk during the mass migration from nests to water[15,16]. Synchronous hatching may also reduce the period during which chemical cues are released by early hatchlings which can attract predators to consume remaining siblings[17]. Turtle embryos can "catch up" to their more developmentally advanced siblings by increasing developmental rates or by hatching prematurely to achieve synchronous hatching[13,18]. Our study suggested a novel mechanism of synchronous hatching. Behavioral thermoregulation enables embryos to select optimal developmental temperature in nests. Within a nest, embryos may move to avoid extreme temperatures for upper-layer eggs experiencing high temperatures, whereas embryos may move to seek heat for lower-layer eggs experience cold temperatures. Therefore, behavioral thermoregulation by embryos probably reduces thermal differentials among eggs within a nest, and thereby synchronizes embryonic development among siblings (Fig. 2).

Embryonic behavioral thermoregulation enhanced the hatching success of turtle eggs in the summer of 2022, during which a severe and prolonged heat wave occurred[19]. This result implies that embryos might thermoregulate behaviorally to avoid lethal temperatures, reducing mortality risk due to extreme heat during development. Further warming experiments are needed to clarify whether this behavior can help buffer embryos from overheating,

**Table 1 Effects of behavioral thermoregulation on embryonic development in different habitats of *Pelodiscus sinensis*.**

| Variable | Thermoregulation | Thermoregulation-inhibited | Statistical analysis |
|---|---|---|---|
| Open habitat | | | |
| Incubation period (days) | 54.74 ± 0.09 ($n = 73$) | 57.67 ± 0.27 ($n = 69$) | $F_{1,140} = 110.561$, $p < 0.001$ |
| Coefficient of variation | 0.010 ± 0.001 ($n = 15$) | 0.034 ± 0.003 ($n = 15$) | $F_{1,28} = 48.791$, $p < 0.001$ |
| Variation range (days) | 1.50 ± 0.22 ($n = 15$) | 5.09 ± 0.54 ($n = 15$) | $F_{1,28} = 38.565$, $p < 0.001$ |
| hatching success (%) | 0.73 ± 0.03 ($n = 15$) | 0.64 ± 0.03 ($n = 15$) | $F_{1,28} = 8.290$; $p = 0.012$ |
| Filtered habitat | | | |
| Incubation period (days) | 54.89 ± 0.13 ($n = 76$) | 58.32 ± 0.29 ($n = 68$) | $F_{1,142} = 123.734$, $p < 0.001$ |
| Coefficient of variation | 0.016 ± 0.003 ($n = 15$) | 0.037 ± 0.003 ($n = 15$) | $F_{1,28} = 29.815$, $p < 0.001$ |
| Variation range (days) | 2.24 ± 0.40 ($n = 15$) | 5.79 ± 0.53 ($n = 15$) | $F_{1,28} = 39.663$, $p < 0.001$ |
| hatching success (%) | 0.75 ± 0.03 ($n = 15$) | 0.64 ± 0.02 ($n = 15$) | $F_{1,28} = 10.678$; $p = 0.006$ |
| Fully-shaded habitat | | | |
| Incubation period (days) | 55.12 ± 0.15 ($n = 66$) | 58.20 ± 0.22 ($n = 66$) | $F_{1,130} = 128.562$, $p < 0.001$ |
| Coefficient of variation | 0.017 ± 0.003 ($n = 15$) | 0.027 ± 0.003 ($n = 15$) | $F_{1,28} = 48.791$, $p = 0.017$ |
| Variation range (days) | 2.32 ± 0.37 ($n = 15$) | 4.08 ± 0.44 ($n = 15$) | $F_{1,28} = 9.293$, $p = 0.005$ |
| hatching success (%) | 0.68 ± 0.04 ($n = 15$) | 0.61 ± 0.01 ($n = 15$) | $F_{1,28} = 3.289$; $p = 0.080$ |

Effects of behavioral thermoregulation on embryonic development were analyzed by generalized linear mixed models with the clutch number as the random factor. Data are presented with means ± SE.

and the importance of this behavior for successful development of embryos exposed to ongoing climate warming. However, our previous study demonstrated that behavioral thermoregulation by embryos did not affect hatching success in the Chinese three-keeled pond turtle (*Mauremys reevesii*)[11]. This may be due to between-species differences in thermal sensitivity of developmental success or between-study differences in incubation environments.

Embryos are one of the most fragile stages of an animal's life history. As such embryos are hypothesized to have developed diverse adaptive strategies to cope with environmental fluctuations given that many species have survived through dramatic environmental changes during earth's history. Our study provides empirical evidence that behavioral thermoregulation can improve developmental success of turtle embryos via shortening the incubation period, enhancing hatching success and optimizing hatching synchronization. These adaptive functions of behavioral thermoregulation may also be applicable to other oviparous species. Behavioral thermoregulation may also enable embryos to develop under optimal thermal conditions, improving offspring quality and fitness[4,11]. Taken together, these findings suggest that turtle embryos should be considered as organisms in their own right, capable of manipulating aspects of their own developmental experience. Future studies are warranted to investigate the generality of our results in other reptile species.

## Materials and Methods

**Study species**. The Chinese softshell turtle (*Pelodiscus sinensis*) is widely distributed in China and usually inhabits rivers or ponds. Adult females lay 2–5 clutches of eggs per annum in a nest (76–100 mm deep) close to a water body from May to August. Each clutch contains 8–30 spherical eggs with a diameter of 20–30 mm and weight of 2–6 g[20]. Wild populations of this species are threatened by overharvesting and habitat destruction, but many populations exist in commercial turtle farms[21]. Incubation temperature affects embryonic development and hatchling traits such as morphology, locomotor performance, growth rate and survival, but does not affect offspring sex[21–23]. In addition, embryos of this species can move inside their eggs to seek a suitable thermal environment, similar to the thermoregulatory behavior seen in adult reptiles[4,5].

**Effects of capsazepine on embryonic development and hatchling traits**. Capsazepine can inhibit embryonic thermoregulatory behavior by blocking the temperature sensors that can detect temperature changes in environments (e.g., TRPV1)[11,12]. To investigate whether capsazepine affected embryonic development and hatchling traits, we collected 15 clutches (mean clutch size = 14) of recently-laid eggs (mean mass = 5.056 g) from a turtle farm at Hebei province of China in July 2021, which is within the natural range of the soft-shelled turtle. We numbered each egg with a pencil in the order of their excavation. All eggs were kept in moist vermiculite (−220 kPa) and transferred to our laboratory in Beijing within one day

of collection. Eggs from the same clutch were half-buried with one another in close contact in boxes (120 × 120 x 40 mm) filled with moistened vermiculite (−220 kPa). Half of the eggs of each clutch were assigned as the capsazepine group (treated with capsazepine) and the other half as the control group (treated with solvent only). We ensured that each egg was viable for the experiment using a cold light to check for embryos, and incubated the boxes containing eggs at a constant temperature of 30 °C in an incubator (KB240; Binder, Germany). Distilled water was added to vermiculite at weekly intervals to ensure that water potential was maintained. When embryos reached developmental stage 11 (8–9 days of incubation), we treated each egg of the capsazepine group with 5 μL of a solution containing 0.37 μg/μL of capsazepine (CAS No. 138977-28-3, $C_{19}H_{21}ClN_2O_2S$, MedChem-Express, China); the same volume of solvent without capsazepine was applied to each egg of the control group. The solvent was composed of 10% anhydrous ethanol, 10% Tween80 and 80% saline. We dropped the solution on the egg surface. After the solution diffused into eggs and the egg surface dried, we put all eggs back to incubation boxes. We applied the solution to eggs three times with an interval of one week.

During the later stages of incubation (24 stages, 37 days) and up until the time all eggs hatched, we observed the eggs every 8 h to record the timing of shell breakage. After hatching, hatchlings were promptly marked on their dorsal carapace for individual identity. Within 24 h after hatching, we measured the mass (± 0.001 g) and the carapace length and width (± 0.01 mm) of the hatchlings. In addition, since the righting time of hatchlings is an important indicator of its neuromuscular developmental performance[13], we turned the hatchlings upside-down on an open platform (250 × 200 × 40 mm), in a room with constant temperature of 28 °C, five times in succession. We recorded the righting trials with a video camera (DCR-SR220E; Sony, Japan) to measure the time required for an individual to successfully right itself within 10 min on five occasions.

**Effects of behavioral thermoregulation on embryonic development and hatchling traits**. In July 2021, we measured the vertical depth (mean depth ± SE = 10.91 ± 0.17 cm) of 70 newly excavated nests by female Chinese soft-shelled turtles at the turtle farm at Hebei province using a digital caliper (CD-6" CSX, Mitutoyo, Japan). To measure the effect of behavioral thermoregulation on embryos and hatchlings, we collected 73 clutches (mean clutch size = 14.6) of recently laid eggs (mean mass = 4.962 g) from the same commercial turtle farm in May 2022. We numbered each egg with a pencil in the order of their excavation. All eggs were subsequently placed in moist vermiculite (−220 kPa) and transferred to the semi-natural experimental site in Zhoushan, Zhejiang Province of China (29.99 N, 122.27E) within one day of collection.

We constructed a total of 73 artificial nests to house the eggs. The mean depth of artificial nests was 10.86 cm, which is similar to the depth of nests constructed by female turtles that we recorded in 2021 (mean depth = 10.91). These nests were evenly distributed at a distance of more than 20 cm from each other across an experimental site (3 × 24 m) with grasses, shrubs and trees to create varying degrees of shading, simulating the oviposition site of female Chinese soft-shelled turtles in their natural habitat[20]. We assigned half of the eggs from each clutch to the behavioral thermoregulation-inhibited group treated with capsazepine, and the other half to the thermoregulation group treated with solvent only. These eggs were placed in three layers within the artificial nest with 4 to 6 eggs in each layer, depending on clutch size. Thermal data-loggers (iButton, DS1921G; MAXIM Integrated Products Ltd., USA) were placed at the surface and 30 cm above the ground of the nest to record soil surface and air temperatures of open, filtered, or fully-shaded habitats (depending on vegetation density). Thermal data-loggers

were also placed at the bottom, middle and top of three nests from each habitat to record nest temperatures hourly.

Out of the 73 clutches, 28 clutches of eggs were used to investigate the difference in developmental stages of embryos between the thermoregulation and thermoregulation-inhibited groups during incubation. We carefully excavated these 28 clutches every 7 days, removing one egg each from each clutch in each experimental group. The excavated eggs were washed with water, then the eggshells were cracked open and the contents poured into dissecting dishes filled with 1× PBS solution (Solarbio, Beijing, China). The developmental stage of embryos was identified using a dissecting microscope on the basis of the morphology of embryos at different developmental stages[24]. The remaining 45 clutches were used to investigate the effect of behavioral thermoregulation on embryonic development and offspring traits. When embryos reached developmental stage of 11–12 (based on observations from the above developmental stage experiment), we carefully excavated all eggs from nests, and applied capsazepine solution to the eggs of the behavioral thermoregulation-inhibited group and solvent treatment without capsazepine to the eggs of the behavioral thermoregulation group. We dropped the solution on the egg surface. After the solution diffused into eggs and the egg surface dried, we put all eggs back to the original nests. We applied the solution to eggs three times with an interval of one week.

When the embryos had reached stage 24 of development (later than the developmental window for behavioral thermoregulation by turtle embryos), we excavated all eggs in nests and identified viable eggs with a cold light. All viable eggs were transferred to our laboratory at Beijing, where each clutch of eggs was clustered together and half buried in moistened vermiculite (−220 kPa) within a transparent box (120 × 120 ×120 mm). We separated the eggs of the thermoregulation group from those of the thermoregulation-inhibited group with a soft polyester mesh in the middle of each box; all eggs within each group were placed so that they were all in contact with each other. All boxes containing eggs from each clutch were incubated in an incubator (KB240; Binder, Germany) at a constant temperature of 30 °C (close to the nest temperature at the end of incubation under semi-natural conditions). We set up live cameras over these transparent boxes to monitor hatching behavior (i.e., eggshell breakage). As soon as each eggshell broke, we recorded the timing of eggshell breakage, and measured hatchling body mass (± 0.001 g). We adopted the method in the constant temperature experiment above to measure each hatchlings' righting time. One hatchling from each treatment did not right itself within the time limit, its data were excluded from subsequent statistical analysis.

**Statistics and reproducibility**. All data were analyzed by IBM SPSS Statistics Version 26.0 (IBM Corp) and tested for normality and homogeneity of variance prior to analysis. Non-parametric tests were used when data did not meet the requirements of parametric testing. Significant differences were determined at $p < 0.05$ and all data are expressed as mean ± SE. We used a generalized linear mixed model to evaluate the effect of capsazepine on incubation period, hatching success and hatchling righting time with treatment as a fixed factor and clutch number as a random factor. A linear mixed model was used to evaluate the effect of capsazepine on the carapace length and width and body mass of hatchlings with treatment as a fixed factor, initial egg mass as a covariate, and clutch number as a random factor. For the semi-natural experiment, we used the Kruskal-Wallis test to compare between-site differences in air and soil surface temperatures, and temperature difference within each nest. Mann-Whitney U tests were used to test for the effect of behavioral thermoregulation on developmental stages of embryos. We evaluated the effect of behavioral thermoregulation on incubation period (mean, variation coefficient and variation range within a clutch), hatching success, and hatchling righting time using a generalized linear mixed model with treatment as a fixed factor and clutch number as a random factor. We evaluated the effect of behavioral thermoregulation on hatchling traits using a mixed linear model with treatment as a fixed factor, initial egg mass as a covariate, and clutch number as a random factor.

**Reporting summary**. Further information on research design is available in the Nature Portfolio Reporting Summary linked to this article.

## Data availability
All data needed to evaluate the conclusions are present in the Supplementary Data 1.

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

## Acknowledgements

We thank Hua Ye and Haijian Lin for their assistance in the laboratory and Rick Shine for comments on the draft of manuscript. We are grateful to three anonymous reviewers for their suggestive comments. This work was supported by grants from National Natural Science Foundation of China (32030013, 31821001). Animals. All the animal studies of turtles conformed to the recommendations in the Guide for the Care and Use of Laboratory Animals of Institute of Zoology, Chinese Academy of Sciences. The research was performed under approvals from the Animal Ethics Committee at the Institute of Zoology, Chinese Academy of Sciences (IOZ14001).

## Author contributions

S.L., B.Z., and X.T.G. conducted the experiments; S.L. and W.G.D. analyzed data; S.L. and W.G.D. prepared the manuscript. W.G.D. conceived and supervised the project.

## Competing interests
The authors declare no competing interests.

**Additional information**

