## [Peer Review File · Communications Biology]

Reviewers' comments:

Reviewer #1 (Remarks to the Author):

This is a clear, straightforward study testing the fitness benefits of embryonic thermoregulation in a turtle by pharmacologically blocking it for a subset of embryos reared in semi-natural nests. The core design is solid, well controlled and the analysis seems appropriate. The results are fascinating and likely to be of broad interest. The original report that turtle embryos move within their eggs to thermoregulate generated a lot of excitement and also controversy and skepticism. This is a lovely follow-up study showing that blocking that behavior really does make a difference for them. Embryos whose thermal receptor was blocked (impairing thermoregulation) developed more slowly and hatched later and less synchronously – all of which could reduce their fitness. There is not a direct test of fitness (e.g. following their survival after hatching) but the paper does provide clear evidence that embryonic thermoregulation makes a difference. This is important and I'm not aware of any other study showing these effects in a naturalistic context.

There are a few points of information (methods) that could be clarified and one claim that needs more justification or a more cautious statement (see #2 below), but overall the manuscript reads well.

SPECIFIC COMMENTS

1) L108 – I suggest “reduce” rather than “minimize” – they're only shortening incubation by a few days, so they would still have to face those risks for quite a while.

2) L121 – Clearly they do improve their hatching synchrony, which is likely beneficial. That could be because they are achieving more similar temperatures (minimize thermal differentials) but I'm not convinced you can actually know that from this experiment. This either needs more evidence, or explanation of how you can know that from these data, or else you need to be clearer about what is actually known vs. a possible, hypothesized mechanism to explain those data. I'd also like to know more about the mixed treatments within presumably the same nest. How can you distinguish embryos just moving to warmer places vs. moving to more similar temperatures? Wouldn't that require embryos who could be the warmest to move to a cooler place, and wouldn't they only do that if it was actually too hot for them? And, if they did that, why would they ignore the temperatures of their drugged siblings & only attend to the temperatures of the ones who are also thermoregulating. I'm convinced of the hatching synchrony, but not by your statement of how it's happening.

3) REF 20 – I don't think this is the citation that you meant to include. Check it.

4) Methods – Clarify if (that?) all eggs from a clutch were maintained together in the same nest, i.e. combining drug-treated & vehicle-treated eggs. Were they intermixed or spatially segregated? Can you confirm (how do you know) that the capsazepine does not move/diffuse from egg to egg? The fact that they still thermoregulate suggests it doesn't, but please clarify.

5) L376 – Fig. 1 caption. Missing a word – advanced faster or more rapidly?

Reviewer #2 (Remarks to the Author):

The authors tested for effects of embryo behavioral thermoregulation in turtles using experimental methods that prevent allow the authors to prevent some embryos from thermoregulating. They did these experiments by manipulating eggs in nests in semi-natural conditions with differing levels of sun

exposure. They found that inhibiting embryo thermoregulation led to longer incubation times and lower hatching success compared to embryos allowed to thermoregulate. Furthermore, nests in which thermoregulation was inhibited had greater hatching asynchrony. The authors use a very nice experimental approach, and the results are broadly interesting. Some specific comments are below.

40 I would avoid using the word tiny or anything else denoting size, for some researchers an embryo the size of these turtle embryos would be quite large.

46 It should be noted that arguments against the potential for embryos to thermoregulate are size-dependent. Thermoregulation may be possible and beneficial in relatively large eggs, but not in small eggs. Given the egg size of these turtles, I doubt even the authors of the papers arguing caution about embryo thermoregulation (I am not one of them) would consider the findings of this study surprising.

94 Odd that body size did not differ between groups. The temperature size rule is very robust across taxa.

103 I think it is worth noting in the Discussion that members of this research group did not find embryonic thermoregulation to increase hatching success in another turtle species (Ye et al. 2019). Therefore, the generality of the authors findings is very much yet to be determined.

120 Consider changing the language here. You did not test whether or not embryo behavioral thermoregulation evolved in order to minimize thermal differential among eggs and synchronize hatching. Greater synchronization may simply be a byproduct.

175 Are capsazepine treatments administered only once? And how is it administered: injected into the egg, spread over the surface, some other means?

253 A lot of pair-wise statistical tests were conducted, but I see no evidence of correction for multiple comparisons. The results shown in Figure 1 are a good example. I recommend the authors either correct for multiple comparisons when appropriate and/or apply statistical modeling approaches that don't require so many pairwise tests.

Reviewer #3 (Remarks to the Author):

Liu et al. conducted an experiment whereby they chemically inhibited the ability of some turtle embryos to behaviorally thermoregulate inside of eggs while allowing others to do so, and then tested the hypothesis that there may be an adaptive function of embryo thermoregulation. They found that embryos which thermoregulated had lower mortality, hatched earlier, and had higher hatching synchrony, all of which are presumably adaptive.

This is a straightforward study that creatively applied methods previously developed by these researchers to test a contentious hypothesis. This is a very interesting piece of work and I commend the authors on a job well done.

My only criticisms of this study, which are minor, are that the effect sizes are not huge and offspring survival or subsequent reproductive success (i.e. actual measurements of fitness) were not evaluated. Regardless, I think these results are really important in the field of reptile thermal ecology as most

researchers still assume that embryos are passive with respect to their thermal environment. This study confirms that at least for some species, not only are they not passive, but embryo thermoregulation affects traits that are likely linked with fitness. This is a pretty neat result, and it will force us (me included) to rethink how we view reptile embryos in the context of thermal adaptation, climate change effects, etc.

Line-by-line comments:

Line 57: It is not clear from this brief description how the blocking of ion channels would inhibit behavioral thermoregulation, and while I'm sure you will explain this later on in the paper, I'd have a brief explanation of how this works in your introduction.

Line 106: I am not sure what you mean by "as opposed to premature hatching", here. What is the difference between faster development (which I assume leads to earlier hatching) and premature hatching? Consider rewording this sentence for clarity.

Lines 110-112: Lizards and turtles are not particularly closely related. I think you should mention that the fitness benefits of early hatching in your study species are not known (assuming that is the case).

Lines 120-122: This is worded in a way that suggests that individual embryos can somehow evaluate the thermal differential in the broader nest and then thermoregulate to compensate for that differential. But I do not see how that would be possible. Consider rewording this sentence to reflect that thermoregulatory decisions made by individual embryos has the outcome of lower variance in hatch date (probably by all embryos seeking a common preferred temperature) rather than giving the impression that embryos are actively trying to reduce the variance between themselves and their siblings.

Lines 123-129: I would temper the wording in this paragraph given you did not conduct an experiment whereby some nests were exposed to a heat wave and others were not. Also, it does not appear that you can know whether eggs actually experienced potentially lethal temperatures or not. While this is suggestive that thermoregulation inside eggs might be important during heat waves, this all seems very speculative at present.

Line 140: I do not understand what is meant by "viable organism", here. Do people not consider turtle embryos to be viable organisms? Even if they were previously thought to not use behavior that much, I do not think anyone ever denied that embryos function as organisms.

Line 160: What do you mean by "temperature sensors", here?

Line 258: Typically written as "generalized linear mixed model" not "generalized mixed linear model".

Line 271: See my previous comment.

Behavioral thermoregulation by reptile embryos promotes hatching success and synchronization (COMMSBIO-23-0560-T)

Reviewer #1 (Remarks to the Author):

This is a clear, straightforward study testing the fitness benefits of embryonic thermoregulation in a turtle by pharmacologically blocking it for a subset of embryos reared in semi-natural nests. The core design is solid, well controlled and the analysis seems appropriate. The results are fascinating and likely to be of broad interest. The original report that turtle embryos move within their eggs to thermoregulate generated a lot of excitement and also controversy and skepticism. This is a lovely follow-up study showing that blocking that behavior really does make a difference for them. Embryos whose thermal receptor was blocked (impairing thermoregulation) developed more slowly and hatched later and less synchronously – all of which could reduce their fitness. There is not a direct test of fitness (e.g. following their survival after hatching) but the paper does provide clear evidence that embryonic thermoregulation makes a difference. This is important and I'm not aware of any other study showing these effects in a naturalistic context.

There are a few points of information (methods) that could be clarified and one claim that needs more justification or a more cautious statement (see #2 below), but overall the manuscript reads well.

Response: We appreciate your constructive comments, which will improve the quality of our manuscript. We have responded to your concerns and suggestions as follows.

SPECIFIC COMMENTS

1) L108 – I suggest “reduce” rather than “minimize” – they're only shortening incubation by a few days, so they would still have to face those risks for quite a while.

Response: Thank you very much for your careful review. Following your suggestion, we changed the word “minimize” to “reduce”.

2) L121 – Clearly they do improve their hatching synchrony, which is likely beneficial. That could be because they are achieving more similar temperatures (minimize thermal differentials) but I'm not convinced you can actually know that from this experiment. This either needs more evidence, or explanation of how you can know that from these data, or else you need to be clearer about what is actually known vs. a possible, hypothesized mechanism to explain those data. I'd also like to know more about the mixed treatments within presumably the same nest. How can you distinguish embryos

just moving to warmer places vs. moving to more similar temperatures? Wouldn't that require embryos who could be the warmest to move to a cooler place, and wouldn't they only do that if it was actually too hot for them? And, if they did that, why would they ignore the temperatures of their drugged siblings & only attend to the temperatures of the ones who are also thermoregulating. I'm convinced of the hatching synchrony, but not by your statement of how it's happening.

Response: Thank you for your insightful comments. We agree that we do not have data to say the exact behavioral mechanism of hatching synchrony. Nonetheless, behavioral thermoregulation enables embryos to select optimal developmental temperature in nests. Within a nest, embryos may move to avoid extreme temperatures for upper-layer eggs experiencing high temperatures, whereas embryos may move to seek heat for lower-layer eggs experience cold temperatures. Therefore, behavioral thermoregulation by embryos probably reduces thermal differentials among eggs within a nest, and thereby synchronize embryonic development among siblings. We add this to the discussion (**Lines 124-130**).

3) REF 20 – I don't think this is the citation that you meant to include. Check it.

Response: Thank you very much for your careful review. We did misquote the reference 20, which has now been replaced with the reference of Mu et al.2015 in **Line 164**.

4) Methods – Clarify if (that?) all eggs from a clutch were maintained together in the same nest, i.e. combining drug-treated & vehicle-treated eggs. Were they intermixed or spatially segregated? Can you confirm (how do you know) that the capsaizepine does not move/diffuse from egg to egg? The fact that they still thermoregulate suggests it doesn't, but please clarify.

Response: We excavated the eggs of both groups from the nests and placed them in separate positions from each other and subsequently dropped the solution on the egg surface. We left all eggs in the air until the diffusion of the solution into eggs and the egg surface dried before putting all eggs back to the original nests for further incubation. Therefore, we believe that the diffusion of capsazepine among eggs, if any, has been minimized. We have added this information to methods (**Line, 192-195; 250-253**)

5) L376 – Fig. 1 caption. Missing a word – advanced faster or more rapidly?

Response: Thanks. We changed this sentence in **Line 393** to “From week 5-7, however, developmental stages **advanced faster** in the thermoregulation group than the thermoregulation-inhibited group”.

Reviewer #2 (Remarks to the Author):

The authors tested for effects of embryo behavioral thermoregulation in turtles using experimental methods that prevent allow the authors to prevent some embryos from thermoregulating. They did these experiments by manipulating eggs in nests in semi-natural conditions with differing levels of sun exposure. They found that inhibiting embryo thermoregulation led to longer incubation times and lower hatching success compared to embryos allowed to thermoregulate. Furthermore, nests in which thermoregulation was inhibited had greater hatchling asynchrony. The authors use a very nice experimental approach, and the results are broadly interesting. Some specific comments are below.

Response: We appreciate your constructive comments, which will improve the quality of our manuscript. We have responded to your concerns and suggestions as follows.

40 I would avoid using the word tiny or anything else denoting size, for some researchers an embryo the size of these turtle embryos would be quite large.

Response: Thanks for your suggestion. We have deleted the word “tiny”.

46 It should be noted that arguments against the potential for embryos to thermoregulate are size-dependent. Thermoregulation may be possible and beneficial in relatively large eggs, but not in small eggs. Given the egg size of these turtles, I doubt even the authors of the papers arguing caution about embryo thermoregulation (I am not one of them) would consider the findings of this study surprising.

Response: Thank you for this insightful suggestion. We agree with you. Our previous study has demonstrated that behavioral thermoregulation is possible in turtles with relatively large eggs including the studied species in this paper, but not in lizards with small eggs. Therefore, we add a sentence here “Actually, embryonic behavioral thermoregulation may be possible in relatively large eggs (e.g., turtles), but not in small eggs (e.g., lizards) (Li 2014)” (**Lines 46-48**)

94 Odd that body size did not differ between groups. The temperature size rule is very robust across taxa.

Response: Thank you for raising this issue. A previous study demonstrated that hatchling body size was not affected by incubation temperature ranging from 24 to 34 °C (Du & Ji. 2003, Journal of Thermal Biology). The temperature difference between treatments induced by behavioral thermoregulation should be much narrow than this wide range of 10 °C important point. Therefore, it is reasonable that behavioral thermoregulation by embryos did not affect hatchling sizes in this species.

103 I think it is worth noting in the Discussion that members of this research group did not find embryonic thermoregulation to increase hatching success in another turtle species (Ye et al. 2019). Therefore, the generality of the authors findings is very much yet to be determined.

Response: Thank you for your suggestion. We have added the following sentences to the Discussion “However, our previous study demonstrated that behavioral thermoregulation by embryos did not affect hatching success in the Chinese three-keeled pond turtle (*Mauremys reevesii*)(Ye et al. 2019). This may be due to between-species differences in thermal sensitivity of developmental success or between-study differences in incubation environments.”(Lines 138-142).

120 Consider changing the language here. You did not test whether or not embryo behavioral thermoregulation evolved in order to minimize thermal differential among eggs and synchronize hatching. Greater synchronization may simply be a byproduct.

Response: We agree with you and have revised the relevant statement here to “Our study suggested a novel mechanism of synchronous hatching. Behavioral thermoregulation enables embryos to select optimal developmental temperature in nests. Within a nest, embryos may move to avoid extreme temperatures for upper-layer eggs experiencing high temperatures, whereas embryos may move to seek heat for lower-layer eggs experience cold temperatures. Therefore, behavioral thermoregulation by embryos probably reduces thermal differentials among eggs within a nest, and thereby synchronize embryonic development among siblings” (Lines 124-131).

175 Are capsazepine treatments administered only once? And how is it administered: injected into the egg, spread over the surface, some other means?

Response: To ensure the effectiveness of the inhibitory effect, we determined the dose and frequency of applied treatments based on pre-experiments and the study by Ye et al. 2019. We applied capsazepine three times with an interval of one week. Specifically, we excavated the eggs of both groups from the nests and placed them in separate positions from each other and subsequently dropped the solution on the egg surface. We left all eggs in the air until the diffusion of the solution into eggs and the egg surface dried before putting all eggs back to the original nests. We have added this information to methods (Lines 192-195; 250-253)

253 A lot of pair-wise statistical tests were conducted, but I see no evidence of correction for multiple comparisons. The results shown in Figure 1 are a good example. I recommend the authors either correct for multiple comparisons when appropriate and/or apply statistical modeling approaches that don’t require so many pairwise tests.

We greatly value your suggestions, but prefer to do pair-wise statistical tests (Mann-Whitney U tests) rather than multiple comparisons (e.g., Kruskal-Wallis test) for the following reasons. To investigate the effect of embryonic behavioral thermoregulation on developmental rates, we conducted six separate sampling sessions on two groups of embryos. Each sampling episode was independent and had no effect on next episode. In addition, we aim to compare the between-treatment differences in developmental stages in each sampling time rather the temporal change in developmental rate. Therefore, we think Mann-Whitney U test is more proper to analyze the data than Kruskal-Wallis test in our case. We are sorry if it confused you because we combined the results of the six separate tests into Figure 1.

Reviewer #3 (Remarks to the Author):

Liu et al. conducted an experiment whereby they chemically inhibited the ability of some turtle embryos to behaviorally thermoregulate inside of eggs while allowing others to do so, and then tested the hypothesis that there may be an adaptive function of embryo thermoregulation. They found that embryos which thermoregulated had lower mortality, hatched earlier, and had higher hatching synchrony, all of which are presumably adaptive.

This is a straightforward study that creatively applied methods previously developed by these researchers to test a contentious hypothesis. This is a very interesting piece of work and I commend the authors on a job well done.

My only criticisms of this study, which are minor, are that the effect sizes are not huge and offspring survival or subsequent reproductive success (i.e. actual measurements of fitness) were not evaluated. Regardless, I think these results are really important in the field of reptile thermal ecology as most researchers still assume that embryos are passive with respect to their thermal environment. This study confirms that at least for some species, not only are they not passive, but embryo thermoregulation affects traits that are likely linked with fitness. This is a pretty neat result, and it will force us (me included) to rethink how we view reptile embryos in the context of thermal adaptation, climate change effects, etc.

Response: We appreciate your constructive comments, which will improve the quality of our manuscript. Additionally, as you suggested, we are doing a follow-up study that addresses the effect of embryonic behavioral thermoregulation on offspring survival and reproductive success.

Line-by-line comments:

Line 57: It is not clear from this brief description how the blocking of ion channels

would inhibit behavioral thermoregulation, and while I'm sure you will explain this later on in the paper, I'd have a brief explanation of how this works in your introduction.

Response: Thank you for suggestion. We now add a brief explanation “pharmacologically blocking transient receptor potential channels (TRPs) that sense temperature changes” here. **(Lines 59-60)**

Line 106: I am not sure what you mean by “as opposed to premature hatching”, here. What is the difference between faster development (which I assume leads to earlier hatching) and premature hatching? Consider rewording this sentence for clarity.

Response: Premature hatching means early hatching as an undeveloped hatchling **(Lines 109-110)**. For example, In the study by Colbert et al. 2010, researchers found that synchronous hatching caused a premature hatching of embryos at a cost of a reduction in hatchling righting capacity.

Lines 110-112: Lizards and turtles are not particularly closely related. I think you should mention that the fitness benefits of early hatching in your study species are not known (assuming that is the case).

Response: Thank you for the suggestion. We add “but such fitness benefits of early hatching are unknown in turtles” to this sentence. **(Line 116)**

Lines 120-122: This is worded in a way that suggests that individual embryos can somehow evaluate the thermal differential in the broader nest and then thermoregulate to compensate for that differential. But I do not see how that would be possible. Consider rewording this sentence to reflect that thermoregulatory decisions made by individual embryos has the outcome of lower variance in hatch date (probably by all embryos seeking a common preferred temperature) rather than giving the impression that embryos are actively trying to reduce the variance between themselves and their siblings.

Response: We agree with you and have rewritten this sentence. “Behavioral thermoregulation enables embryos to select optimal developmental temperature in nests. Within a nest, embryos may move to avoid extreme temperatures for upper-layer eggs experiencing high temperatures, whereas embryos may move to seek heat for lower-layer eggs experience cold temperatures. Therefore, behavioral thermoregulation by embryos probably reduces thermal differentials among eggs within a nest, and thereby synchronize embryonic development among siblings” **(Lines 124-131)**

Lines 123-129: I would temper the wording in this paragraph given you did not conduct an experiment whereby some nests were exposed to a heat wave and others were not.

Also, it does not appear that you can know whether eggs actually experienced potentially lethal temperatures or not. While this is suggestive that thermoregulation inside eggs might be important during heat waves, this all seems very speculative at present.

Response: We agree and have modified the paragraph as you suggested. “This result implies that embryos might thermoregulate behaviorally to avoid lethal temperatures, reducing mortality risk due to extreme heat during development. Further warming experiments are needed to clarify whether this behavior can help buffer embryos from overheating, and the importance of this behavior for successful development of embryos exposed to ongoing climate warming.” (Lines 134-138)

Line 140: I do not understand what is meant by “viable organism”, here. Do people not consider turtle embryos to be viable organisms? Even if they were previously thought to not use behavior that much, I do not think anyone ever denied that embryos function as organisms.

Response: We agree and delete “viable”.

Line 160: What do you mean by “temperature sensors”, here?

Response: We add some words to explain temperature sensors. “temperature sensors that can detect temperature changes in environments” (Line 173).

Line 258: Typically written as “generalized linear mixed model” not “generalized mixed linear model”.

Response: Changed as you suggested.

Line 271: See my previous comment.

Response: Changed as you suggested.

REVIEWERS' COMMENTS:

Reviewer #1 (Remarks to the Author):

The authors have responded appropriately to all of my comments, improving the manuscript. From my perspective, this is a very nice and interesting study (as mentioned in my first review) and the issues with overreach in interpretation and lack of clarity have been fixed.

I will just recommend a couple of small word changes:

Line 110: Premature hatchlings are not "undeveloped" – they are just less developed, i.e., at an earlier developmental stage. (Undeveloped eggs cannot hatch.) Please fix this.

Line 111: Shorter or shortening (not shorten)

Reviewer #2 (Remarks to the Author):

The authors have done a very nice job addressing my comments, and I appreciate the effort they put into making revisions. I have no further comments.

Reviewer #3 (Remarks to the Author):

The authors did an excellent job responding to my feedback. I have no further concerns and congratulate them on a really fascinating study.

**Behavioral thermoregulation by reptile embryos promotes hatching
success and synchronization (COMMSBIO-23-0560-A)**

Dear Editor:

Thank you for giving us the opportunity to revise the manuscript again. We have responded reviewers' concerns and suggestions below.

REVIEWERS' COMMENTS:

Reviewer #1 (Remarks to the Author):

The authors have responded appropriately to all of my comments, improving the manuscript. From my perspective, this is a very nice and interesting study (as mentioned in my first review) and the issues with overreach in interpretation and lack of clarity have been fixed.

I will just recommend a couple of small word changes:

Line 110: Premature hatchlings are not “undeveloped” – they are just less developed, i.e., at an earlier developmental stage. (Undeveloped eggs cannot hatch.) Please fix this.

Response: Following your suggestion, we've changed the explanation here to “hatching at an earlier developmental stage”.

Line 111: Shorter or shortening (not shorten)

Response: Thank you very much for your careful review. Following your suggestion, we changed the word “shorten” to “Shortening”.

Reviewer #2 (Remarks to the Author):

The authors have done a very nice job addressing my comments, and I appreciate the effort they put into making revisions. I have no further comments.

Response: We greatly appreciate your acknowledgement of our research, and we value the suggestions and concerns you have raised.

Reviewer #3 (Remarks to the Author):

The authors did an excellent job responding to my feedback. I have no further concerns and congratulate them on a really fascinating study.

Response: We are very grateful for your valuable suggestions and positive comments on our work!